# Enhancement of Sensitivity with High−Reflective−Index Guided−Wave Nanomaterials for a Long−Range Surface Plasmon Resonance Sensor

**DOI:** 10.3390/nano12010168

**Published:** 2022-01-04

**Authors:** Leiming Wu, Kai Che, Yuanjiang Xiang, Yuwen Qin

**Affiliations:** 1Institute of Advanced Photonics Technology, School of Information Engineering, Guangdong University of Technology, Guangzhou 510006, China; leiming_wu@gdut.edu.cn; 2Guangdong Provincial Key Laboratory of Information Photonics Technology, Guangdong University of Technology, Guangzhou 510006, China; 3Southern Marine Science and Engineering Guangdong Laboratory (Zhuhai), Zhuhai 519000, China; 4Department of Electrical and Information Engineering, Hubei University of Automotive Technology, Shiyan 442002, China; 20160010@huat.edu.cn; 5School of Physics and Electronics, Hunan University, Changsha 410082, China

**Keywords:** long−range surface plasmon resonance, high−refractive−index guided−wave layer, sensitivity

## Abstract

A guided−wave long−range surface plasmon resonance (GW−LRSPR) sensor was proposed in this investigation. In the proposed sensor, high−refractive−index (RI) dielectric films (i.e., CH_3_NH_3_PbBr_3_ perovskite, silicon) served as the guided−wave (GW) layer, which was combined with the long−range surface plasmon resonance (LRSPR) structure to form the GW−LRSPR sensing structure. The theoretical results based on the transfer matrix method (TMM) demonstrated that the LRSPR signal was enhanced by the additional high#x2212;RI GW layer, which was called the GW−LRSPR signal. The achieved GW−LRSPR signal had a strong ability to perceive the analyte. By optimizing the low− and high−RI dielectrics in the GW−LRSPR sensing structure, we obtained the highest sensitivity (S) of 1340.4 RIU^−1^ based on a CH_3_NH_3_PbBr_3_ GW layer, and the corresponding figure of merit (FOM) was 8.16 × 10^4^ RIU^−1^ deg^−1^. Compared with the conventional LRSPR sensor (S = 688.9 RIU^−1^), the sensitivity of this new type of sensor was improved by nearly 94%.

## 1. Introduction

The Kretschmann–Raether (KR) configuration [1] is an important carrier for the excitation of the surface plasmon resonance (SPR) phenomenon [2,3,4]. Metals, such as gold (Au), silver (Ag), aluminum (Al), and copper (Cu), contain a large number of free electrons on their surface. These free electrons oscillate up and down in the direction perpendicular to the surface of the metal film under the excitation of TM−polarized light, thereby forming a surface plasmon wave (SPW). When the evanescent wave generated by the attenuated total reflection of the incident light matches the SPW, they will resonate and form a resonance signal. The resonance signal allows for sensitive detection that is used as the principle of designing sensors. The SPR sensors are widely applied in environmental monitoring [5,6], biochemical reactions [7,8], medical diagnosis [9,10], and other fields. Sreekanth et al. [11] proposed a hyperbolic metamaterials−based plasmonic biosensor with a high sensitivity of 30,000 nm per RIU, which can detect ultralow−molecular−weight (244 Da) biomolecules at picomolar concentrations. Garol et al. [12] reported a nanoporous gold metamaterials−based plasmonic sensor, where the sensitivity was demonstrated to be 15,000 nm per RIU. Kabashin et al. [13] used plasmonic nanorod metamaterials to design a biosensor, where the sensitivity was more than 30,000 nm per RIU. Moreover, the plasmonic nanorod metamaterials can also be applied in high−pressure sensors [14,15], as well as surface−enhanced Raman scattering (SERS) [16]. Runowski et al. [14] reported that gold nanorods can be used as a high−pressure sensor to effectively detect phase transitions of compressed compounds. Chiang et al. [15] proposed a compact plasmonic metal–insulator–metal pressure sensor that the maximum sensitivity was as high as 592.44 nm/MPa.

In the SPR configuration, Au is usually selected as the metal layer for exciting SPR because of its strong oxidation resistance. In the resonance signal, there are many sensitive variables, such as the intensity of the reflectance [17,18], the resonance angle [19,20], and the phase [21,22], which can be used to track and detect a change in the sensing analyte. The SPR signal generated by the Au film has a strong detection ability for the analyte close to the sensing interface. However, the detection ability will gradually decay as the detection depth increases. The detection depth of the conventional SPR sensor is generally less than 200 nm [23], which makes it difficult to effectively detect some macromolecules with a diameter of several hundred nanometers. In order to solve this defect, LRSPR [24,25] is used as a detection signal to improve the detection depth of the sensor. In the LRSPR configuration, the metal layer is thin enough so that the upper and lower SPWs of the metal film are coupled together, thereby enhancing the electric field at the sensing interface. A previous report [26] indicated that the detection depth of the LRSPR sensor can exceed 1 μm, which greatly improves the sensor’s ability to detect analytes. Compared with an SPR sensing structure, LRSPR has many advantages [24,27]: (1) stronger electric field intensity, (2) deeper detection distance, (3) narrower resonance detection signal, and (4) higher detection accuracy. In addition to the LRSPR sensor, some other novel sensors were also reported with high performance, such as a plasmonic photonic crystal fiber sensor [28], plasmonic nanorod sensor [29], metal–insulator–metal (MIM) sensor [30], monolithic metal dimer−on−film−structure−based sensor [31], and toroidal−metadevices−based sensor [32]. These innovative sensing structures have all contributed to improving the performance of plasmonic sensors.

In order to improve the sensitivity, a sensing structure, named guided−wave SPR (GWSPR), was proposed, which enhances the sensitivity of conventional sensors by adding an additional GW layer [33,34]. The result demonstrates that the sensitivity of the SPR sensor is improved by one order of magnitude with an additional GW layer. However, the additional GW layer greatly enhances the electric field intensity but does not contribute to the increase of the detection depth. Therefore, in order to enhance the sensitivity and increase the detection depth, a novel sandwich layer of a metal–dielectric−structure−based plasmonic sensor is proposed. TMM, which is a method that is widely used in the design of sensing structures, was used as the theoretical method to investigate the performance of the proposed GW−RSPR sensor. It was found that the resonance signal of the LRSPR configuration was enhanced by an additional high−RI GW layer, which greatly improved the electric field intensity, as well as the sensitivity, and retained the ability of the LRSPR sensor to detect a large depth. With such excellent performance, the GW−LRSPR sensing structure is promising for applications in biosensing, chemical sensing, and optical sensing.

## 2. Design Consideration and Numerical Model

The configuration of an LRSPR sensor is composed of a sandwich structure made up of a low−RI layer/metal film/low−RI layer (Figure 1a). In this study, a GW−LRSPR sensing structure made up of a prism/low−RI dielectric/high#x2212;RI dielectric/Au/sensing analyte was proposed. In our investigation, cytop was selected as the low−RI dielectric layer near the side of the prism, which has a low RI of 1.34. In addition to cytop, other low−RI materials, such as LiF and MgF_2_, are also widely used in the design of LRSPR sensors with high sensitivity [20]. The low−RI layer on the other side of the LRSPR configuration was the sensing analyte, and its RI was n_s_ = 1.33 + Δn_s_. Here, Δn_s_ represents the slight change in the sensing analyte, where this change could be a biological action or a chemical reaction. In the sandwich structure, Au was used as the metal layer to excite the long−range surface plasma waves due to its good oxidation resistance. The RI of Au is expressed by the Drude–Lorentz model as follows [35]:(1)$$n_{Au}={\left(1-\frac{\lambda^{2}\lambda_{C}}{\lambda_{P}^{2}(\lambda_{C}+i\lambda )}\right)}^{1/2}$$
where *λ* = 633 nm, *λ_C_* = 8.9342 × 10^−6^ m, and *λ_P_* = 1.6826 × 10^−7^ m. In addition, BK7, SF10, and 2S2G (Ge_20_Ga_5_Sb_10_S_65_) were used as coupling prisms in the sensing structure. Their refractive indices are defined as:(2)$$n_{Bk7}={\left(\frac{1.03961212\lambda^{2}}{\lambda^{2}-0.00600069867}+\frac{0.231792344\lambda^{2}}{\lambda^{2}-0.0200179144}+\frac{1.03961212\lambda^{2}}{\lambda^{2}-103.560653}+1\right)}^{1/2}$$
for the *BK*7 prism,
(3)$$n_{\mathrm{SF}10}={\left(\frac{1.62153902\lambda^{2}}{\lambda^{2}-0.0122241457}+\frac{0.256287842\lambda^{2}}{\lambda^{2}-0.0595736775}+\frac{1.64447552\lambda^{2}}{\lambda^{2}-147.468793}+1\right)}^{1/2}$$
for the SF10 prism, and
(4)$$n_{2S2G}=2.24047+\frac{2.693\times 10^{-2}}{\lambda^{2}}+\frac{8.08\times 10^{-3}}{\lambda^{4}}$$
for the 2*S*2*G* prism. When the wavelength of incident light is 633 nm, the RI of the *BK*7, SF10, and 2S2G prisms are 1.5151, 1.723, and 2.358, respectively. To improve the sensitivity of the LRSPR sensor, we considered inserting a GW material with high RI between the Au film and the low−RI dielectric layer to form a GW−LRSPR sensing structure (Figure 1b). The GW material (high#x2212;RI dielectric film) was CH_3_NH_3_PbBr_3_ perovskite, where its RI is 2 + 0.003i at λ = 633 nm [36,37]. The parameters optimization of the sensing structure is discussed in Section 3 “Results.” The results showed that the optimal thicknesses of the low−RI dielectric, high#x2212;RI dielectric, and Au film were 2430 nm, 4 nm, and 10 nm, respectively.

In the proposed sensor, the low−RI dielectric film was cytop, which is widely used in sensing applications and can be deposited on the cover glass using evaporation. Thereafter, the high−RI dielectric film of perovskite could be transferred to the surface of the cytop using a self−assembly method. Next, the gold film was sprayed onto the surface of the high−RI dielectric film by the method of magnetic co−sputtering. Finally, the cover glass and the prism were glued together with the matching liquid to prepare the proposed sensing structure. The sensing process was divided into three steps: first, fasten the sample cell tightly on the surface of the sensing structure; second, use a pump to control the analyte so that it flows slowly through the sample cell; finally, use the resonance signal to detect the analyte flowing through the sample cell.

In a multi−layer sensing structure, the transfer matrix method (TMM) [38,39] is usually used as the theoretical method to study the performance of a proposed plasmon sensor. The simulation software used was Matlab. Each layer in the multi−layer structure can be expressed as a matrix:(5)$$M_{k}=\left[\begin{array}{l} \cos\beta_{k}\left(-i\sin\beta_{k}\right)/q_{k}\\ -iq_{k}\sin\beta_{k}\cos\beta_{k} \end{array}\right]$$
where $\beta_{k}=\frac{2\pi d_{k}}{\lambda }{\left(\varepsilon_{k}-n_{1}^{2}\sin^{2}\theta \right)}^{1/2}$ and $q_{k}=\frac{{\left(\varepsilon_{k}-n_{1}^{2}\sin^{2}\theta \right)}^{1/2}}{\varepsilon_{k}}$. Here, *d_k_* is the thickness of each layer in the multilayer sensing structure, *ε_k_* is the dielectric constant of each layer, and *n_1_* is the RI of the prism. The characteristic transfer matrix of the sensing configuration is as follows:(6)$$M=\prod_{k=2}^{N-1}M_{k}=\left[\begin{array}{l} M_{11}\;M_{12}\\ M_{21}\;M_{22} \end{array}\right]$$

The reflectance of the multilayer sensing structure is given as:(7)$$R={\left|r_{p}\right|}^{2}={\left|\frac{\left(M_{11}+M_{12}q_{N}\right)q_{1}-\left(M_{21}+M_{22}q_{N}\right)}{\left(M_{11}+M_{12}q_{N}\right)q_{1}+\left(M_{21}+M_{22}q_{N}\right)}\right|}^{2}$$

The intensity sensitivity (*S*) and figure of merit (*FOM*) are expressed as:(8)$$S=\frac{\Delta R}{\Delta n_{s}}$$
and
(9)$$FOM=\frac{S}{FWHM}$$
where *FWHM* is the full width at half maximum and Δ*R* is the intensity change of the reflectance curve caused by the change in the sensing analyte (Δ*n_s_*). In our numerical calculations, *Δ**n_s_* represents a slight change in the sensing medium and was set to 0.0001.

## 3. Results

The resonance signals excited from the configuration of the LRSPR and GW−LRSPR are shown in Figure 1c, which indicates that an additional high−RI GW layer (CH_3_NH_3_PbBr_3_ perovskite) could improve the quality of the resonance signal. We first calculated the maximum loss and then calculated half of its maximum. At the half−maximum point on the SPR curve, two different angle values were obtained. Then, by subtracting the lesser angle from the greater angle, the difference obtained was the FWHM. The result demonstrated that the GW−LRSPR had a narrower FWHM and a deeper resonance dip than that of the LRSPR. In contrast, the resonance signal of the GW−LRSPR was stronger than that of the LRSPR. According to Equation (8), the corresponding intensity sensitivities of the LRSPR and GW−LRSPR were calculated using the TMM (Figure 1b). The result showed that the GW−LRSPR sensor had a higher sensitivity than that of the LRSPR sensor due to the enhancement of the resonance signal by the GW layer. After adding the GW layer, the sensitivity of the LRSPR sensor increased from 688.9 RIU^−1^ to 1107.8 RIU^−1^. Furthermore, the FWHM of the resonance signal became narrower after adding the GW layer, which promoted the improvement of the FOM according to Equation (9). The FOM of the LRSPR sensor was 4.53 × 10^4^ RIU^−1^ deg^−1^, which was improved to be 8.58 × 10^4^ RIU^−1^ deg^−1^ after adding a high−RI GW layer to form the GW−LRSPR configuration. For the resonance signals, the stronger the resonance, the deeper the dip. In the GW−LRSPR sensor, the thickness of both the high#x2212;RI dielectric and Au film had a great influence on the resonance signal. The area between the two white curves in Figure 1e is the strong resonance region of the proposed sensor at different d_2_ and d_Au_ values. Therefore, in the design of the sensing structure, the selection of the thickness of both the high−RI dielectric and Au film needed to correspond to the strong resonance region. Figure 1f shows the detection result of the resonance signal on the change in the RI of the sensing analyte based on the GW−LRSPR sensor. It was found that the resonance signal was sensitive to the change in RI, and it gradually moved to a larger angle as the RI of the sensing analyte increased.

In order to clearly demonstrate the enhancement of the high−RI GW layer on the sensitivity of the LRSPR sensor, the electric field distribution comparison between the LRSPR and GW−LRSPR is discussed. The electric field distribution curves of the LRSPR and GW−LRSPR at their resonance angles are shown in Figure 2. The results of the comparison showed that the electric field intensity of the sensor structure was improved by the additional high−RI GW layer. The enhancement of the electric field at the sensing interface improved the sensor’s ability to perceive changes in the sensing medium, thereby increasing the sensitivity of the sensor.

There are four approaches for calculating the sensitivity of a sensor [23], including intensity sensitivity, angular sensitivity, spectral sensitivity, and phase sensitivity. For angular sensitivity, a previous report demonstrated that the selection of coupling prism has a great impact on the performance of the sensor [19]. Therefore, it is necessary to analyze the influence of the coupling prism on the sensitivity of the proposed GW−LRSPR sensor. Herein, the intensity sensitivity was used to calculate and analyze the performance of the proposed sensor, which is different from the angular sensitivity. Figure 3a,b shows the reflectance and intensity sensitivity for the proposed GW−LRSPR sensor based on SF10 and BK7, respectively. The intensity sensitivity for the SF10-based GW−LRSPR sensor was 1131.2 RIU^−1^ (Figure 3a), while the intensity sensitivity for the BK7−based GW−LRSPR sensor was 1030.1 RIU^−1^ (Figure 3b). These results were similar to the intensity sensitivity of the 2S2G−based sensor shown in Figure 1d,f. Assuming that the RI of the prism was variable, the intensity sensitivities of the proposed GW−LRSPR sensor under different prisms were calculated and are shown in Figure 3c. The results demonstrated that the coupling prism with different RIs had little effect on the intensity sensitivity of the proposed GW−LRSPR sensor. Therefore, 2S2G, SF10, and BK7 can all be used as coupling prisms in the proposed GW−LRSPR sensor, and will not have a significant impact on intensity sensitivity. According to Snell’s law, a change in the RI of the prism will lead to a change in the critical angle of total reflection. With the increase in the RI of the prism, the critical angle of total reflection will gradually decrease; therefore, the resonance signal will also move toward the direction of a small angle. In this process, the characteristics of the resonance signal were basically unchanged and the sensitivity changed slightly.

For the following discussion, the SF10 was selected as the coupling prism in the sensing structure. After completing the selection of the coupling prism, we optimized the thickness of the low−RI dielectric layer and the high−RI GW layer to obtain the optimal sensing sensitivity. The enhancement mechanism is that the surface plasmon on the metal surface is allowed to be transmitted to the high−RI GW film, thereby enhancing the electric field at the sensing interface. When the high−RI GW film is thin enough, most of the evanescent field is located in the analyte area, increasing the detection range and sensitivity [33]. Figure 4a shows the intensity sensitivity as a function of the thickness of the low−RI dielectric film (d_1_). When the d_1_ ranged from 1000 nm to 3800 nm, the intensity sensitivity gradually increased first and reached its maximum value of S_Max_ = 1326.6 RIU^−1^ at d_1_ = 2430 nm. The FOM corresponding to the maximum sensitivity was 8.1 × 10^4^ RIU^−1^ deg^−1^. Furthermore, when n_1_ changed from 1.33 to 1.36, the corresponding sensitivity also changed, and a greater sensitivity could be obtained in the range of 1.33 to 1.34 (Figure 4b). Therefore, the cytop material, which has an n_1_ of 1.34, was selected as the low−RI dielectric film, and the thickness was defined as 2430 nm in the proposed GW−LRSPR sensor. Additionally, the thickness of the high−RI GW layer was also an important parameter to optimize the performance of the GW−LRSPR sensor. Figure 4c shows the resonance signals excited from the proposed GW−LRSPR sensor at different d_2_ values. The corresponding intensity sensitivity and FOM were achieved and are shown in Figure 4d. The result showed that the highest sensitivity (S = 1340.4 RIU^−1^) was obtained at d_2_ = 4 nm, with FOM = 8.16 × 10^4^ RIU^−1^ deg^−1^. Therefore, in order to maintain a high sensitivity (S > 1200 RIU^−1^) for the proposed GW−LRSPR sensor, the allowable thickness variation range for the high−RI dielectric film was 2100 nm to 2800 nm, and the allowable thickness variation range for the low−RI dielectric film was 2 nm to 6 nm.

The resonance signal excited from the GW−LRSPR sensor was sensitive to the change in RI in the sensing analyte. A slight change in the sensing analyte can cause an obvious shift in the resonance angle (Figure 5a). When the RI of the sensing analyte changed from 1.32 to 1.345, the FWHM of the resonance signal became narrower and narrower (Figure 5b). After calculations, the results showed that the intensity sensitivity of the GW−LRSPR sensor increased to the highest value (S = 1533.7 RIU^−1^) at n_s_ = 1.334 and then gradually decreased (Figure 5c). Furthermore, the FOM of the GW−LRSPR sensor continued to increase in the range of n_s_ = 1.32 ~ 1.345 and reach the maximum value (FOM = 11.45 × 10^4^ RIU^−1^ deg^−1^) at n_s_ = 1.337 (Figure 5d).

In addition to CH_3_NH_3_PbBr_3_ perovskite, other high−RI dielectrics can also be used as GW layers to improve the sensing sensitivity of an LRSPR sensor. Figure 6a shows the intensity sensitivity optimized by n_2_ and d_2_. The result indicated that the GW dielectrics with different high RIs could achieve higher sensitivities in the appropriate thickness range. Herein, silicon was exemplified as another high−RI GW layer to enhance the sensitivity of the GW−LRSPR sensor. The resonance signal and intensity sensitivity of the GW−LRSPR sensor with a 3 nm thick silicon GW layer is shown in Figure 6b. The result demonstrated that high sensitivity of 1392.3 RIU^−1^ could be obtained when the GW layer was silicon. The above results demonstrated that the silicon−f the sensing analyte changed from 1.330 to 1.335, the corresponding resonance signal moved significantly (Figure 6c).

Furthermore, the sensitivity of the proposed GW−LRSPR sensor could also be calculated in terms of the spectral sensitivity (*S* = *Δ**λ/Δn_s_*). When the n_s_ changed from 1.33 to 1.3301 (Δn_s_ = 0.0001), the spectral sensitivity was calculated to be 7097 nm/RIU. The variation in the resonance signal with respect to the wavelength is shown in Figure 7a. It was found that the proposed sensor was sensitive to the change in n_s_, and even if the n_s_ had a change of only 0.0001, it could still be effectively detected. The spectral sensitivity and FOM under different n_s_ values are shown in Figure 7b. When the n_s_ increased from 1.33 to 1.34, both the S and FOM changed accordingly. After calculations, the highest spectral sensitivities could be obtained as 7285 nm/RIU at n_s_ = 1.335. Such excellent performance means that the GW−LRSPR sensing structure can play an active role in many fields, including environmental monitoring, medical diagnosis, and biological testing.

## 4. Conclusions

In this study, a high−RI GW layer (i.e., CH_3_NH_3_PbBr_3_ perovskite, silicon) was combined with an LRSPR structure to form the proposed GW−LRSPR sensing structure. It was found that the resonance signal of the LRSPR structure was enhanced by the additional high−RI GW layer. When the GW layer was introduced into the LRSPR structure, the electric field at the sensing interface was effectively enhanced, thereby improving the sensing ability of the sensor. By optimizing the GW−LRSPR sensing structure, the maximum sensitivity (1340.4 RIU^−1^) was obtained with the GW layer of the CH_3_NH_3_PbBr_3_ perovskite, and the corresponding FOM was 8.16 × 10^4^ RIU^−1^ deg^−1^. The GW−LRSPR sensor was sensitive to the change in the sensing analyte. When the RI of the sensing analyte was 1.334, a high sensitivity of S = 1533.7 RIU^−1^ could be obtained. Moreover, another high−RI GW layer of silicon could also be used to improve the sensing sensitivity in the GW−LRSPR structure. The result showed that adding a 3 nm thick silicon guide-wave layer could increase the sensitivity to 1392.3 RIU^−1^.

## Figures and Tables

**Figure 1 nanomaterials-12-00168-f001:**
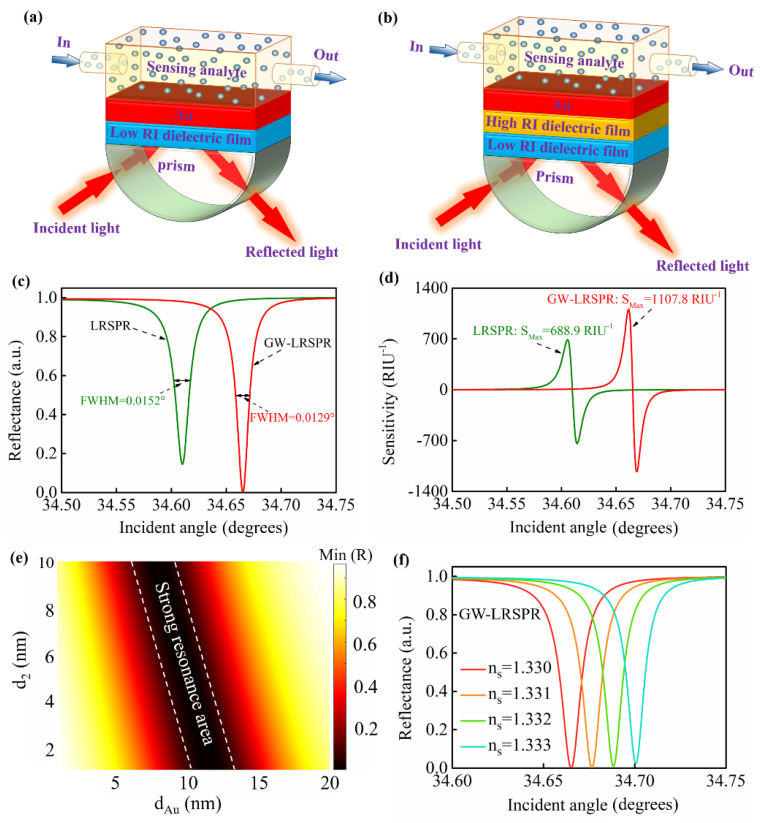
(**a**) Schematic diagram of the 10 nm thick Au−based conventional long−range surface plasmon resonance sensor. (**b**) Schematic diagram of the proposed high−RI−dielectric film−based GW−LRSPR sensor. (**c**) The resonance signals generated from the LRSPR and GW−LRSPR sensors based on the 2S2G prism, where their full width at half maximum (*FWHM*s) were 0.0152° and 0.0129°, respectively. (**d**) The sensitivities of the conventional Au−based LRSPR sensor and the novel GW−LRSPR sensor. (**e**) The strong resonance area of the proposed GW−LRSPR sensor at different d_2_ and d_Au_ values. (**f**) The curves of reflectance excited from the GW−LRSPR sensor when *n_s_* ranged from 1.330 to 1.333. Note: the results of Figure c–f were obtained from the LRSPR and GW−LRSPR configurations based on the 2S2G prism.

**Figure 2 nanomaterials-12-00168-f002:**
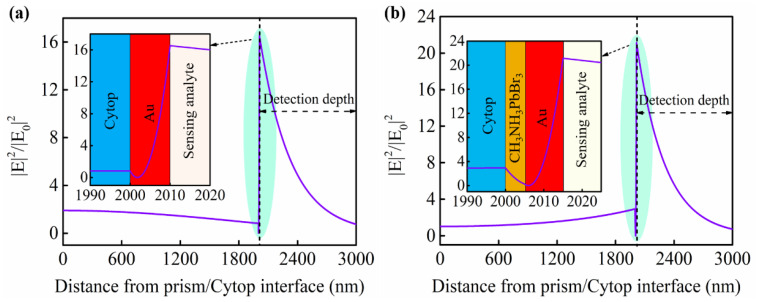
The electric field distribution of the plasmon sensors based on (**a**) LRSPR and (**b**) GW−LRSPR at the resonance angle (*θ_res_* = 34.6100° for LRSPR, *θ_res_* = 34.6652° for GW−LRSPR).

**Figure 3 nanomaterials-12-00168-f003:**
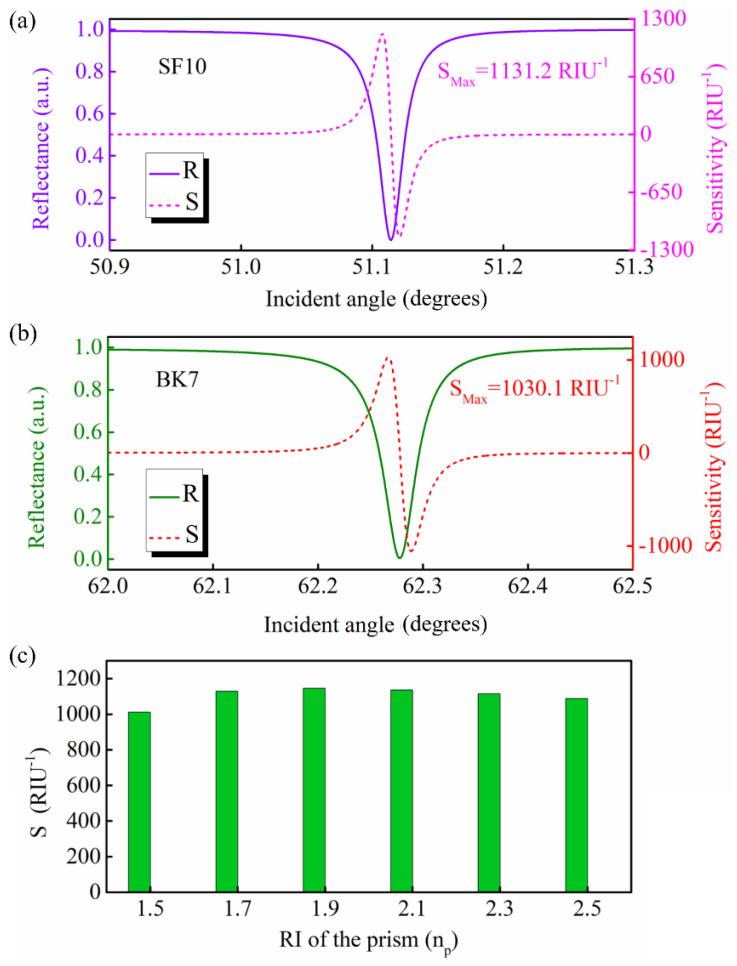
Reflectance and sensitivity of the proposed GW−LRSPR sensor based on (**a**) SF10 and (**b**) BK7 prisms. (**c**) The sensitivity of the proposed GW−LRSPR sensor based on different prisms.

**Figure 4 nanomaterials-12-00168-f004:**
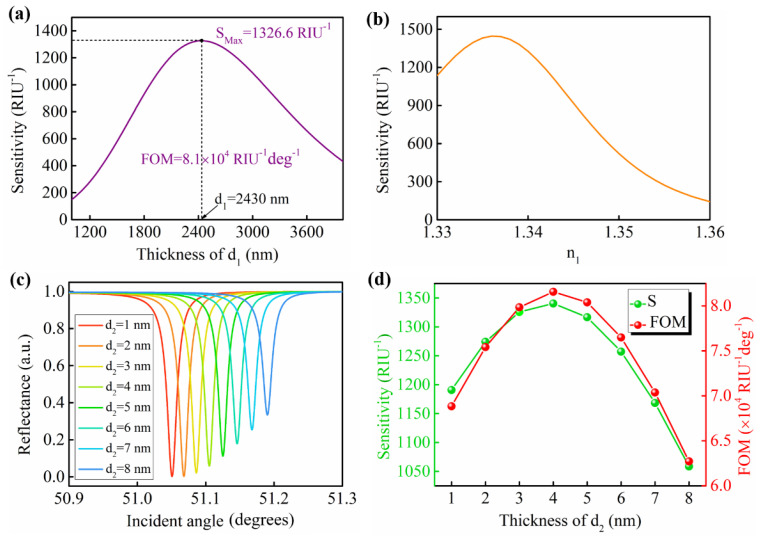
(**a**) Variation of the intensity sensitivity when the thickness of the low−RI dielectric film (d_1_) ranged from 1000 nm to 3800 nm for the proposed GW−LRSPR sensor. (**b**) The intensity sensitivity as a function of the RI of the low−RI dielectric film (n_1_) at d_1_ = 2430 nm. (**c**) The reflectance curves of the proposed GW−LRSPR sensor at different thicknesses of the low−RI GW layer (d_2_) when d_1_ = 2430 nm and n_1_ = 1.34. (**d**) The intensity sensitivity and FOM of the GW−LRSPR sensor at different d_2_ values.

**Figure 5 nanomaterials-12-00168-f005:**
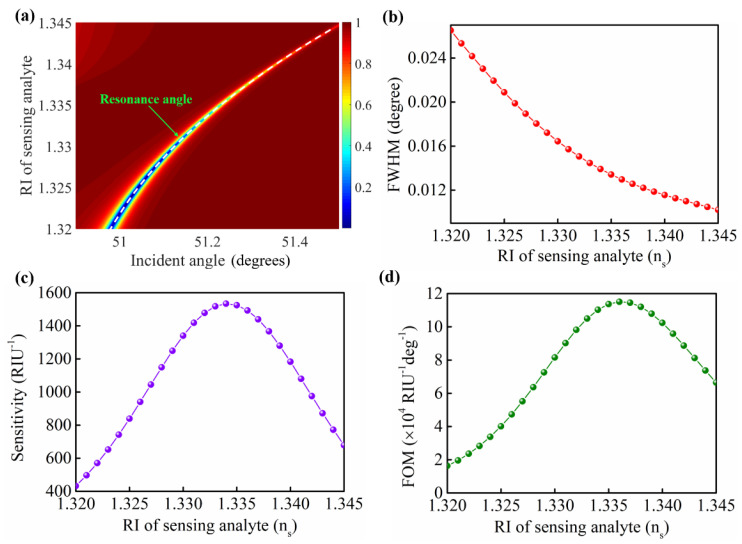
(**a**) The resonance angle shift caused by the RI change of the sensing analyte. (**b**) The FWHM of the resonance signal under different RIs of the sensing analytes. (**c**) The intensity sensitivity of the proposed GW−LRSPR sensor with different n_s_ values. (**d**) The FOM of the proposed GW−LRSPR sensor with different n_s_ values.

**Figure 6 nanomaterials-12-00168-f006:**
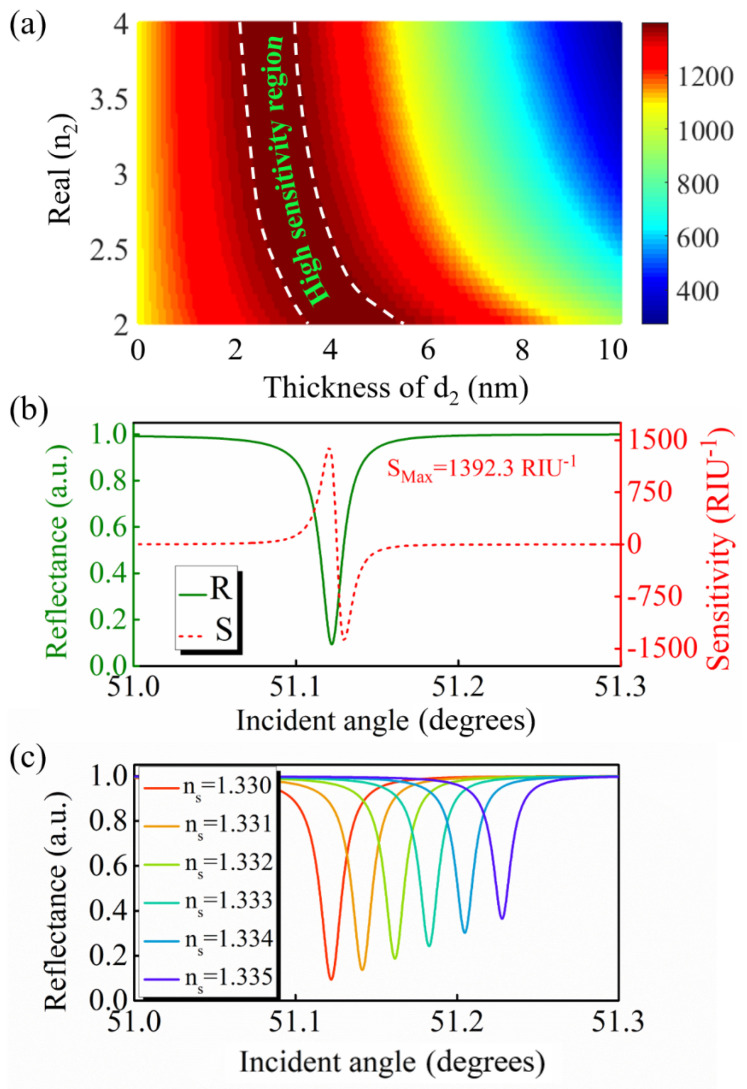
(**a**) Optimizing n_2_ and d_2_ to obtain the highest sensitivity. (**b**) The resonance signal of the proposed GW−LRSPR sensor and its corresponding sensitivity at d_2_ = 3 nm when the high−RI GW layer was silicon. (**c**) Shift of the resonance signal at different RIs of the sensing analytes.

**Figure 7 nanomaterials-12-00168-f007:**
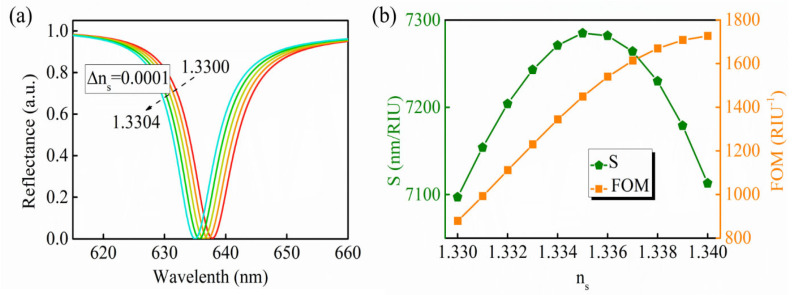
(**a**) The variation of the resonance signal with respect to the wavelength when the n_s_ ranged from 1.33 to 1.3304 (Δn_s_ = 0.0001). (**b**) The spectral sensitivity (*S* = *Δ**λ/Δn_s_*) and the corresponding FOM of the proposed GW−LRSPR based on silicon. Note: the incident angle was fixed at 51.12°.

## Data Availability

The data presented in this study are available on request from the corresponding author.

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
