# Peer review of "Enhancement of Sensitivity with High−Reflective−Index Guided−Wave Nanomaterials for a Long−Range Surface Plasmon Resonance Sensor"

_nanomaterials, 2022, doi:10.3390/nano12010168_

Round 1

Reviewer 1 Report

The paper under consideration does not meet the standard of the Nanomaterials journal. The paper is full of technical errors and lacks information which makes it unattractive and inappropriate to be published. The paper is numerically based yet there is no single piece of information about the software, method, model, etc explained in the paper. Moreover, the sensitivity is wrongly calculated and the formula is incorrect. How do sensitivity and Q-factor have the same unit? Sensitivity is related to a change in intensity or angle with respect to the change in ambient index (delta intensity/delta refractive index). 

There is no such novelty in the paper that deserves appreciation and is considered for publication. The author has introduced a high index film between the gold and low index film.  After section 2, the author has jumped to the Results and discussion section without giving the detailed geometric parameters optimization. 

Based on my above-mentioned concerns, I am not willing to accept this paper. 

Author Response

1. The paper under consideration does not meet the standard of the Nanomaterials journal. The paper is full of technical errors and lacks information which makes it unattractive and inappropriate to be published. The paper is numerically based yet there is no single piece of information about the software, method, model, etc explained in the paper.

Answer: Thanks for your professional comments. According to the reviewer’s comment, we revised the manuscript. The information about the software, method, model are shown in the section 2. 

2. Moreover, the sensitivity is wrongly calculated and the formula is incorrect. How do sensitivity and Q-factor have the same unit? Sensitivity is related to a change in intensity or angle with respect to the change in ambient index (delta intensity/delta refractive index).

Answer: Thanks for your professional comments. According to the reviewer’s comment, we revised the manuscript. The sensitivity calculated in our manuscript is the intensity sensitivity (Laser Photonics Rev., 2011, 5(4), 571–606). ΔR is the intensity change of Reflectance curve caused by the change in sensing analyte (Δns). The “Q factor” in our manuscript is revised to “figure of merit (FOM)” [OSA Continuum, 2020, 3(8), 2253-2263], and it is defined as FOM=S/FWHM, Where FWHM is the full width at half height. The unit is RIU-1 deg-1

 3. There is no such novelty in the paper that deserves appreciation and is considered for publication. The author has introduced a high index film between the gold and low index film.  After section 2, the author has jumped to the Results and discussion section without giving the detailed geometric parameters optimization.

Answer: Thanks for your professional comments. The parameters optimization of the sensing structure is discussed in section 3 “Results”. The results show that the optimal thickness of the low RI dielectric, high RI dielectric, and Au film are 2430 nm, 4 nm, and 10 nm, respectively. The optimized parameters have been added after section 2.

Reviewer 2 Report

In this manuscript, the authors proposed a high RI GW layer (e.g., CH3NH3PbBr3 perovskite, silicon) which is combined with LRSPR structure to form the proposed GW-LRSPR sensing structure. They claim that the highest sensitivity (S) is 1326.6 RIU-1, and the corresponding quality (Q) factor is 1166.5 RIU-1. Technically, the authors employ the TMM method with apparent authority, and the results seem valid. However, the necessary elucidation of the physical mechanism and references are absent in the manuscript to explain the obtained results. Therefore, in my opinion, a major revision is needed to accommodate the high-quality requirements of this Journal. 

  1. The theoretical method (i.e., TMM) used in this work should be mentioned in the abstract and introduction section.
  2. The novelty of this work is vague. Many plasmonic sensors have been published with better frequency responses. This work proposed a sandwich layer of the metal-dielectric structure-based plasmonic sensor. In the introduction section, the authors should mention the novelty of the designed structure and comment on the other approaches of plasmonic sensors, e.g., plasmonic photonic crystal fiber sensor (see OSA Continuum 3, 2253-2263 (2020)), plasmonic nanorod sensing structure (see Nanomaterials, 2019, 9, 1691), and metal-insulator-metal (MIM) sensors (see Opt. Express, 29, 32364-32376 (2021)).
  3. In line 113, the definition of the Q factor may be a wrong one. Equation (9) should be “figure of merit (FOM)” (see J. Phys. D: Appl. Phys., 2020,53, 115401). The Q factor has a different definition. Would you please check throughout this manuscript?
  4. To check the accuracy of sensitivity (S) and quality (Q) factor. In the text, please quantitatively describe the number of dR, dθ and FWHM.
  5. Please brief describe the influence of the thickness of sandwich layers (i.e., Au, high RI dielectric film and low RI dielectric film) on optical properties. The range of workable layer thickness is suggested providing in the text.
  6. The fabrication issue of the proposed structure should be briefly mentioned and cited in the relevant articles.
  7. Please briefly clarify why the incident wavelength of 633 nm is used in this work. Equations (2), (3) and (4) should be a fixed number if the incident wavelength of 633 nm is used.
  8. Please clarify how to inject the samples into the detection area?
  9. In Fig. 3 and Fig. 4 (d), why the sensitivity and Q factor share the same unit. Please check them.
  10. The most important thing is that there is almost no discussion of any physical mechanism in the manuscript except some brief description of the superficial phenomenon. Please describe in more detail the mechanism of the results obtained from Figs. 3-5.

Author Response

1. The theoretical method (i.e., TMM) used in this work should be mentioned in the abstract and introduction section.

Answer: Thanks for your professional comments. The TMM method is described and added in the abstract and introduction section.

2. The novelty of this work is vague. Many plasmonic sensors have been published with better frequency responses. This work proposed a sandwich layer of the metal-dielectric structure-based plasmonic sensor. In the introduction section, the authors should mention the novelty of the designed structure and comment on the other approaches of plasmonic sensors, e.g., plasmonic photonic crystal fiber sensor (see OSA Continuum 3, 2253-2263 (2020)), plasmonic nanorod sensing structure (see Nanomaterials, 2019, 9, 1691), and metal-insulator-metal (MIM) sensors (see Opt. Express, 29, 32364-32376 (2021)).

Answer: Thanks to the reviewer for recommending these novel sensing structures, and the corresponding descriptions and discussions have been added to the manuscript.

3. In line 113, the definition of the Q factor may be a wrong one. Equation (9) should be “figure of merit (FOM)” (see J. Phys. D: Appl. Phys., 2020,53, 115401). The Q factor has a different definition. Would you please check throughout this manuscript?

Answer: Thanks for your professional comments. The “Q factor” in our manuscript is revised to “figure of merit (FOM)”.

4. To check the accuracy of sensitivity (S) and quality (Q) factor. In the text, please quantitatively describe the number of dR, dθ and FWHM.

Answer: Thanks for your professional comments. Where FWHM is the full width at half height, and dR is the intensity change of Reflectance curve caused by the change in sensing analyte (dns). In our numerical calculations, dns represents a slight change in the sensing medium and is set to 0.0001.

5. Please brief describe the influence of the thickness of sandwich layers (i.e., Au, high RI dielectric film and low RI dielectric film) on optical properties. The range of workable layer thickness is suggested providing in the text.

Answer: Thanks for your professional comments. As the outermost layer of the sensing structure, the Au film is most vulnerable to external damage, so its thickness should not be too thin, and we set its thickness to 10 nm. In order to maintain high sensitivity (S > 1200 RIU-1), the allowable thickness variation range for high RI dielectric film is 2100 nm to 2800 nm, and the allowable thickness variation range for low RI dielectric film is 2 nm to 6 nm.

6. The fabrication issue of the proposed structure should be briefly mentioned and cited in the relevant articles.

Answer: Thanks for your professional comments. In the proposed sensor, the low RI dielectric film is cytop, which is widely used in sensing applications and can be deposited on the cover glass by evaporation. Thereafter, the high RI dielectric film of perovskite can be transferred to the surface of cytop by self-assembly method. Next, the gold film is sprayed onto the surface of the high RI dielectric film by the method of magnetic co-sputtering. Finally, the cover glass and the prism are glued together with the matching liquid to prepare the proposed sensing structure.

7. Please briefly clarify why the incident wavelength of 633 nm is used in this work. Equations (2), (3) and (4) should be a fixed number if the incident wavelength of 633 nm is used.

Answer: Thanks for your professional comments. He-Ne laser is a common light source, which is widely used in sensing technology. Its wavelength is 633 nanometers and the spot quality is very good, so we use it as the incident light source in the theoretical calculations. When the wavelength of incident light is 633 nm, the RI of BK7, SF10, and 2S2G prisms are 1.5151, 1.723, and 2.358, respectively.

8. Please clarify how to inject the samples into the detection area?

Answer: Thanks for your professional comments. The sensing process can be divided into three steps: first, fasten the sample cell tightly on the surface of the sensing structure; secondly, use a pump to control the analyte to flow slowly through the sample cell; finally, use the resonance signal to detect the analyte flowing through the sample cell.

9. In Fig. 3 and Fig. 4 (d), why the sensitivity and Q factor share the same unit. Please check them.

Answer: Thanks for your professional comments. The “Q factor” is revised to “figure of merit (FOM)”, and the unit is RIU-1 deg-1.

10. The most important thing is that there is almost no discussion of any physical mechanism in the manuscript except some brief description of the superficial phenomenon. Please describe in more detail the mechanism of the results obtained from Figs. 3-5.

Answer: Thanks for your professional comments. Some physical mechanisms of the proposed sensing structure were discussed in the section of Results.

Reviewer 3 Report

This paper discuss via numerical simulations a model for a long-range SPR sensor. The structure comprises an Au film embedded in dielectric layers on a coupling prism.

the authors report analyses on the sensitivity and quality factor looking at different parameters such as the used prism material and the thickness of the layers. 

The topic is not new at all, and during the last 20 years tons of papers have published on SPR sensors and on long-range SPR configurations.

The results reported here seems to demonstrate some improvement in sensitivity, but a good comparison with the state-of-the-art is missing. In particular the authors cite 34 papers, the largest part very old (before 2010). I strongly recommend a radical revision of the discussion considering an up-to-date state-of-the-art analysis. Some papers on the topic are, for example :doi.org/10.3390/s18092757;   doi.org/10.1007/s11468-018-0814-3;  doi.org/10.1016/j.optlaseng.2018.09.013;   doi.org/10.1364/OE.27.009550;   doi.org/10.1063/1.4921200;

but many others can be considered. In particular it is not interesting at all to mention papers from more than 10 years ago (5 years are already enough). The authors should also introduce the topic of SPR sensors considering the top performers in term of sensitivity (experimental works!): 

doi.org/10.1038/nmat2546

doi.org/10.1039/C9NH00168A

doi.org/10.1038/nmat4609

Another important aspect to discuss is the use of perovskite as dielectric layer. Perovskite is typically photoluminescent materials, and to me the choice to us it sounds weird. Why not to use another dielectric material? maybe also easier to be deposited with a controlled thickness?

minor correction: figure has a very bad quality! it needs to be improved.

The manuscript can be reconsidered only after a significant improvement.

Author Response

1. This paper discuss via numerical simulations a model for a long-range SPR sensor. The structure comprises an Au film embedded in dielectric layers on a coupling prism. The authors report analyses on the sensitivity and quality factor looking at different parameters such as the used prism material and the thickness of the layers. The topic is not new at all, and during the last 20 years tons of papers have published on SPR sensors and on long-range SPR configurations.

Answer: Thanks for your professional comments. In the past 20 years, the principles of SPR and LRSPR have been widely used to design sensors. The innovation of our research is to improve the LRSPR structure by adding an additional high RI guided-wave layer. The result shows that the sensitivity of the proposed GW-LRSPR sensor has been greatly improved by the additional high RI guided-wave layer.

2. The results reported here seems to demonstrate some improvement in sensitivity, but a good comparison with the state-of-the-art is missing. In particular the authors cite 34 papers, the largest part very old (before 2010). I strongly recommend a radical revision of the discussion considering an up-to-date state-of-the-art analysis. Some papers on the topic are, for example : doi.org/10.3390/s18092757;

doi.org/10.1007/s11468-018-0814-3;

doi.org/10.1016/j.optlaseng.2018.09.013; 

doi.org/10.1364/OE.27.009550; 

doi.org/10.1063/1.4921200;

but many others can be considered. In particular it is not interesting at all to mention papers from more than 10 years ago (5 years are already enough). The authors should also introduce the topic of SPR sensors considering the top performers in term of sensitivity (experimental works!): 

doi.org/10.1038/nmat2546

doi.org/10.1039/C9NH00168A

doi.org/10.1038/nmat4609

Answer: Thank you very much for recommending these excellent research results to us. We have introduced and cited these references in the manuscript.

3. Another important aspect to discuss is the use of perovskite as dielectric layer. Perovskite is typically photoluminescent materials, and to me the choice to us it sounds weird. Why not to use another dielectric material? maybe also easier to be deposited with a controlled thickness?

Answer: Thanks for your professional comments. In the proposed sensing structure, the high RI guided-wave layer need to meet the condition: the real part of the RI must be large enough, and the loss must be small. The RI of CH3NH3PbBr3 perovskite meet the condition, which is used to design the sensor in our manuscript. Other materials, such as silicon, can also be used as high RI guided-wave layer to enhance sensing sensitivity in the proposed sensor. The result of using silicon as a guided-wave layer to enhance sensitivity is shown in Figure 6.

4. minor correction: figure has a very bad quality! It needs to be improved.

Answer: Thanks for your professional comments. We have improved the pixels of the figures in the manuscript to make the figures look clearer.

5. The manuscript can be reconsidered only after a significant improvement.

Answer: Thanks for your professional comments. We have made major revision to the manuscript based on the reviewer’s suggestions to make it closer to the requirements of publication.

Reviewer 4 Report

In this work, “Enhancement of sensitivity with high reflective index guided-wave layer for long-range surface plasmon resonance sensor”, the authors propose a GW-LRSPR sensor. In the proposed sensor, high-RI dielectric films are served as a guide wave layer which are combined with long-range surface plasmon resonance structure to form the sensing mechanism. Based on the extracted results, the authors claimed that the sensitivity of this mechanism is improved by 90%, comparing to a conventional one. Overall, this manuscript has a strong potential for a second review after applying the issues and addressing the shortcomings listed below:

1-The authors should polish/revise some grammatical mistakes and typos along the manuscript. I invite the authors to read their manuscript carefully and make the required changes where necessary.

2-In the Introduction section, while discussing the recent developments in the field of plasmonic sensing, the following works should be considered and cited to give a more general view to the possible readers of the work: [(i) Monolithic metal dimer-on-film structure: new plasmonic properties introduced by the underlying metal, Nano Letters 20, 2087-2093 (2020); (ii) Terahertz plasmonics: the rise of toroidal metadevices towards immunobiosensings, Materials Today 32, 108-130 (2020)].

3-Please explain how did you calculate FWHM values in Figure 1.

4-For the schematics in Figure 1, please explain the effect of thickness of Au layer and high RI dielectric film to the overall sensing performance of the proposed platform.

5-What is the operation region (Vis, NIR, MIR, etc.) of the proposed platform? Please explain.

Author Response

In this work, “Enhancement of sensitivity with high reflective index guided-wave layer for long-range surface plasmon resonance sensor”, the authors propose a GW-LRSPR sensor. In the proposed sensor, high-RI dielectric films are served as a guide wave layer which are combined with long-range surface plasmon resonance structure to form the sensing mechanism. Based on the extracted results, the authors claimed that the sensitivity of this mechanism is improved by 90%, comparing to a conventional one. Overall, this manuscript has a strong potential for a second review after applying the issues and addressing the shortcomings listed below:

1-The authors should polish/revise some grammatical mistakes and typos along the manuscript. I invite the authors to read their manuscript carefully and make the required changes where necessary.

Answer: Thanks for your professional comments. We carefully checked the manuscript and revised the errors.

2-In the Introduction section, while discussing the recent developments in the field of plasmonic sensing, the following works should be considered and cited to give a more general view to the possible readers of the work: [(i) Monolithic metal dimer-on-film structure: new plasmonic properties introduced by the underlying metal, Nano Letters 20, 2087-2093 (2020); (ii) Terahertz plasmonics: the rise of toroidal metadevices towards immunobiosensings, Materials Today 32, 108-130 (2020)].

Answer: Thank you very much for recommending these excellent research results to us. We have introduced and cited these references in the manuscript.

3-Please explain how did you calculate FWHM values in Figure 1.

Answer: Thanks for your professional comments. We first calculate the maximum loss, and then calculate half of its maximum. At the half-height point on the SPR curve, two different points on angle are obtained. Then, subtract these two angles, and the difference obtained is the FWHM. The explanation has been added to the manuscript.

4-For the schematics in Figure 1, please explain the effect of thickness of Au layer and high RI dielectric film to the overall sensing performance of the proposed platform.

Answer: Thanks for your professional comments. For resonance signals, the stronger the resonance causes the deeper the dip. In the GW-LRSPR sensor, the thickness of both the high RI dielectric and Au film has a great influence on the resonance signal. The area between the two white curves in Figure 1(e) is the strong resonance region of the proposed sensor at different d2 and dAu. Therefore, in the design of the sensing structure, the selection of the thickness of both the high RI dielectric and Au film needs to correspond to the strong resonance region.

5-What is the operation region (Vis, NIR, MIR, etc.) of the proposed platform? Please explain.

Answer: Thanks for your professional comments. In the proposed sensing structure, the high RI guided-wave layer need to meet the condition: the real part of the RI must be large enough, and the loss must be small. Previous report [J. Phys. Chem. C, 2016, 120(1), 616–620] indicated that the RI of CH3NH3PbBr3 perovskite meet the condition at λ = 550 nm ~ 1100 nm. In the proposed sensor, CH3NH3PbBr3 perovskite is used as the guided-wave layer to enhance the sensitivity. Therefore, the operation region of the proposed sensing structure at Vis ~ NIR region.

Round 2

Reviewer 1 Report

I am still not sure about the unit of FOM. How unit of FOM is 1/RIU.deg? Explain and make corrections throughout the paper.  I would also suggest the author calculate the sensitivity with respect to the change in resonance wavelength (S=nm/RIU). Because in other papers, the authors have mentioned the sensitivity in terms of amplitude and wavelength interrogation methods.  

Author Response

1.  I am still not sure about the unit of FOM. How unit of FOM is 1/RIU.deg? Explain and make corrections throughout the paper.  I would also suggest the author calculate the sensitivity with respect to the change in resonance wavelength (S=nm/RIU). Because in other papers, the authors have mentioned the sensitivity in terms of amplitude and wavelength interrogation methods.

Answer: Thanks for your professional comments. The sensitivity calculated in our manuscript is intensity sensitivity, represents the change in reflectance per unit change in refractive index of sensing medium. The reflectance is a dimensionless quantity, so the unit of sensitivity is 1/RIU. The FOM is defined as FOM = S/FWHM, where FWHM is the full width at half height, and its unit is degree. Therefore, the unit of FOM is RIU-1 deg-1. According to the reviewer’s comment, we have also calculated the sensitivity with respect to the change in resonance wavelength (S=nm/RIU). The result is shown in Figure 7. When the ns changes from 1.33 to 1.3301 (Δns = 0.0001), the spectral sensitivity is calculated to be 7097 nm/RIU. The variation of resonance signal with respect to the wavelength is shown in Figure 7(a). It is found that the proposed sensor is sensitive to the change in ns, and even if the ns has a change of only 0.0001, it can still be effectively detected. The spectral sensitivity and FOM under different ns are shown in Figure 7(b). With the ns increases from 1.33 to 1.34, both the S and FOM will change accordingly. After calculation, the highest spectral sensitivity can be obtained as 7285 nm/RIU at ns = 1.335.

Reviewer 2 Report

Authors have addressed the comments. Therefore, revised manuscript can be accepted for the publication. 

Author Response

Thank the reviewer for their professional review of our manuscript.

Reviewer 3 Report

The authors have improved the manuscript according to the received comments. 

To me the paper can be accepted now.

Author Response

The authors have improved the manuscript according to the received comments. To me the paper can be accepted now.

Answer:Thank the reviewer for their professional review of our manuscript.

Reviewer 4 Report

In its current form, the revised manuscript is suitable for publication.

Author Response

1. In its current form, the revised manuscript is suitable for publication.

Answer:Thank the reviewer for their professional review of our manuscript.